# The Role of 'Digitalization' in German Sustainability Bank Reporting

**Florian Diener \*** and **Miroslav Špaček**

Department of Entrepreneurship, Faculty of Business Administration, University of Economics in Prague, Churchillovo nám. 4, 137 00 Praha 31, Czech Republic; miroslav.spacek@vse.cz
**\*** Correspondence: dief00@vse.cz; Tel.: +49-(0)151-4053-2978

**Abstract:** The financial services sector, particularly with respect to today's banking industry, is aiming to make a digital transition. Sustainable reporting is a holistic new reporting approach in banking and has only become partially mandatory for the sector. Thus, this paper makes a contribution to the current analysis approach and further development of the German Sustainability Code as well as associated legal approaches. It concerns the assessment of mandatory sustainable reporting in the light of constantly changing market conditions and stricter legal requirements for stakeholder data responsibility. In specific, it focuses on a digital evolving business environment and is intended to provide an insight into the perception of the topic of digitalization in the banking sector. The assessment is based on the structure of the German Sustainability Code. Based on 113 bank reports, a multiple regression analysis of 1410 codings of the keyword 'digital' is carried out. The results show that banks partly and not fully address digital issues in their reporting. It transpires that the emphasis is on seven criteria, while social elements are totally ignored. The paper shows a structural inequality within sustainable bank reporting with regard to digitalization. It also shows that issues are not adequately addressed and covered in legal reporting standards and that the provision of information to stakeholders on specific issues is largely undefined.

**Keywords:** bank; digital; digitalization; financial services; innovation; strategy; sustainability; reporting

**JEL Classification:** G20; G21; O30; O33; Q56

## 1. Introduction

In recent years, sustainable reporting has expanded (on a voluntary basis). Many companies nowadays formulate annual sustainability reports, and there is a wide array of standards that determine content and validity. Sustainability reporting approaches can also be seen as synonymous with other non-financial reporting terms, such as corporate social responsibility or triple bottom line reporting. It functions as an element of holistic reporting and is a further development of existing reporting approaches that combines the analysis of financial and non-financial performance in one report.

The main objective of sustainability reporting is to maintain and develop confidence in companies in order to create a sustainable and fully informed environment, in particular because company decisions have a direct impact on markets and stakeholders. Consequently, labour organisations, (civil) society, and citizens are affected as well as the level of trust they have. Sustainable aspects are rarely based on financial information alone. They consist of 'an assessment of risks and opportunities using information on a wide variety of immediate and future issues' (Global Reporting Initiative 2019). By publishing sustainable reports, it ensures that companies consider important issues within their general responsibility. This reporting approach enables them to transfer information on risks and opportunities in the context of sustainability to the public. An overall increased transparency leads to

a better understanding of business relationships, responsibility, and decision making, which helps to maintain and build trust in companies.

With regard to the banking crisis in 2007, there was a lack of sufficient and holistic reporting and regulation standards in the context of the banking system and insufficient reporting with regard to sustainability, for which there is an increased need for information at a strategic and at an operational level (Barth and Landsman 2010). However, until then, there were no reporting standards or approaches that promoted sustainable reporting and informed customers and the market holistically about a bank's sustainability approaches. This knowledge deficit has been addressed by appropriate developments, in particular in sustainability reporting standards and related approaches, but it has still not been examined whether the standards developed today do meet these information efforts and will lead to the desired result.

As an indispensable necessity, banking sector also had to adapt to these market changes, which led to a quick development of reporting approaches. In order to satisfy the information needs of stakeholders, banks are now addressing the various problems of a sustainable economy in a variety of ways. Social issues, market developments, as well as technological topics such as digitalization, which are of great importance for banks today, are being discussed in the paper. In parallel, debates have risen at the European level concerning the character and extent of corporate reporting on social, environmental, and governance problems and one of the most significant drivers of today's economy: the megatrend digitalization. These technological developments in particular are leading to fundamental changes in financial services, while at the same time triggering developments that may be of systemic importance. Due to increasing social digitalization and the growing importance of technology for the general public, banks are also forced to undergo a process of digital adaptation. On the basis of 26 interviews we conducted with German bank managers and specialists, it became evident that digitalization is a key issue for banks, which should be approached at different levels and from different perspectives. Even if reporting on digitalization in sustainability reports is voluntary, the interviews indicated that the topic was important for banks and should be addressed in a holistic way. So, the results of the interviews identified it as an elementary component of current and future banking business. However, as traditional companies and their industries adapt slowly and ineffectively to the modern changing markets, there is a high risk of disruption caused by new technologies and business models. If such changes are missed by (system-relevant) financial institutions such as banks, then financial services and the whole economic system will be endangered. Changing customer needs and the growing influence of technologies and new banking providers on the market (FinTechs) are forcing a fundamental reorientation of banks' traditional business models and their processes and products, so that banks run even more the risk of being disrupted if they do not adapt. In this context, it is very important to report on this development in its entirety.

One first official approach to close this lack of information and to fulfil transparent thoughts was a directive on extending the financial reporting on non-financial and diversity issues enacted by the EU Commission in December 2014 (European Parliament 2014). In September 2016, the Federal Government accepted and introduced to the German Bundestag the proposal of the Corporate Social Responsibility Directive Implementation Act (CSR-RUG) submitted by the Federal Ministry of Justice and Consumer Protection. On March 2017, the Bundestag determinedly counselled and accepted the order after the second/third interpretation, transforming it into German law, in particular the trade codes § 289b et seq. After its promulgation, the law fundamentally came into force. The reporting conditions are determinately influencing selected firms and organisations focussed on capital markets, including credit institutions and insurance firms, even if they are not often registered. The reports are demanded for companies with more than 500 workers (European Parliament 2014)—see also sections 289b et seq. German Commercial Code. If the aforesaid limit value is exceeded, individual services or non-financial aspects, including environmental, employment, and social issues, staff, respect for human rights, battle against corruption and diversity, must be reported to the governing agencies by the businesses involved; however, the main areas of focus are not defined more precisely.

Within this development, as part of its commission and public relations work, the Federal Association of German Banking Association (namely: bankenverband) took part and dealt with the question of sustainability reporting in medium-sized banks and held conversations with different stakeholders. It should be noted at this stage as the German Council for Sustainable Development (RNE[1]). Its duty is to create contributions for the execution of the German Sustainability Strategy, to define certain areas of action and initiatives, and to make sustainability a major government concern (German Council for Sustainable Development 2019), with which profound working contacts have been developed in the last few years. This led to a collaboration contract between the bank association and the RNE, i.e., The Sustainability Code (DNK[2]). The aim of this cooperation is to ensure that medium-sized banks receive the DNK as an easy-to-use instrument of RNE, as these banks will be obliged to report on non-financial elements of their actions due to the large number of employees (German Council for Sustainable Development 2019). Even though the DNK is primarily a reporting approach for the German market, the insights already gained can serve as a guideline and can be transferred to respective markets in other countries and their legislation so that cross-national comparability can be achieved based on a holistic approach. These days, digitalization and sustainability are closely related, as technological development at the corporate level can ensure the long-term existence of a company, especially today (Prashara 2019). This is confirmed by Cybercom (2019) and Stuermer et al. (2017) in the version on 'digital sustainability' and 'digitalization' which is based on the long-term oriented production and further development of digital knowledge. It is 'an approach that harnesses one of the most powerful forces for societal change to deliver what we need and want in a sustainable way. Further, it represents a 21st century tool for discussing, reflecting on, and assessing our real individual and societal needs and wants' (Cybercom 2019, p. 3). Ten prerequisites of digital sustainability describe these (Stuermer et al. 2017).

The first four criteria concern the characteristics of the digital good (elaborateness, transparent structures, sematic data, and distributed location), the other five criteria focus on the characteristics of the ecosystem (open licensing regime, shared tacit knowledge, participatory culture, good governance, and diversified funding), and the last criterion concerns the effects on society (contributing to sustainable development). Based on the concept of sustainability, which has so far been used primarily in connection with ecological issues (Cybercom 2019, p. 8 et seq.), digital sustainability describes the conscious use of resources in such a way that their creation and use today does not compromise the needs of future generations. Consequently, digital approaches are sustainably further developed and thus continuously accelerated and improved. Based on Cybercom (2019, p. 4) and Stuermer et al. (2017), it differs from the original definition of sustainability and refers exclusively to intangible goods, so-called knowledge goods, which are accessible to an ecosystem, can be used by a large number of people, and can also be further developed as an open source. The social effects of digital sustainability in banking include process optimisation and the creation of more transparency through digital technologies. Since digitalization is increasingly influencing the entire economic and market environment, the financial market is also affected by this change and must address and position itself in the context of its importance and responsibility to inform its customers and the market. This would be in line with the digital sustainability approach. Thus, the DNK provides the basis for a holistic view of current developments in the financial market.

All parties, i.e., above all the stakeholders themselves [e.g., investors, individual customers, employees and trade unions, public and industrial sectors, etc.], need and could be fully informed about the changing challenges and transition brought about by digitalization to ensure a high level of transparency in a changing market environment (Lubin and Esty 2014). Transparent reporting begins with detailed positioning on all topics and must be carried out at all levels of the company, especially

---

[1]　In German: Rat für Nachhaltige Entwicklung.
[2]　In German: Deutsche Nachhaltigkeitskodex.

in the case of an all-encompassing topic such as digitalization. Based on the need for legally compliant reporting, the banking market described above and the competitive position as well as the associated changes for the banks must be taken into account within the framework of responsible and sustainable corporate management and the holistic duty to provide information. Special attention must be paid to the topic of digitalization, as it is of particular importance for the current and future banking market. As a result of the Brexit, the German banking system will become increasingly important as, in addition to the French financial sector, it is an important pillar for the European system (Howarth and Quaglia 2018). The high German standards can be seen as an example of high efficiency and reliability and thus as an outstanding example for a banking system that is still in development. Furthermore, this system finances the strongest economy in the EU and is considered as one of the strongest in the world. Due to the accessibility, the available number of reports, and the broad acceptance of the DNK, this paper focuses in particular on the application of the framework in the German banking sector.

This results in two research questions (RQ), which should be answered by qualitative keyword analysis and regression analysis, so that on the one hand insights can be gained into the level of awareness of the topic of digitalization and on the other hand insights into the most important criteria for banks within sustainable reporting.

RQ1: What is the banks' awareness of digitalization within sustainability reporting?
RQ2: What are the criteria of non-financial reporting on which banks focus the most in the context of digitalization?

The paper is structured as follows: First, it focuses on digitalization in the financial sector by looking at the changes in today's banking market and thus confirming the change and pressure caused by the digital evolution. Second, it focuses on the Sustainability Code, which also provides the theoretical framework for the analysis of the work. After a further explanation of the methodology, the results of the qualitative keyword analysis and the multiple regression analysis of the sustainability reports are discussed. Finally, conclusions are drawn, and proposals for future studies in the field of sustainability reporting in the banking sector are discussed.

## 2. Digitalization in the Finance Industry: A Change in Banking

Banking is changing: A recent study demonstrates, for example, the increasing and imminent impact on the financial sector of contemporary business models (Dorfleitner and Hornuf 2016). The danger of disruption by new technologies or competitive business models is high because traditional businesses and their industrial sectors adapt to contemporary evolving economies with slow and ineffective results (Christensen 2006; Christensen and Bower 1996, 1995; Christensen et al. 2002; Christensen and Raynor 2003; Christensen et al. 2015). As suggested by Walter (2016, p. 17), four primary influencing variables determine the business strategies and day-to day practices of the current financial intermediaries, all with three subfactors (Technology: Product and Solutions, Customer Interaction, Process and Info-management/Customer: Convenience, Performance, Price/Market: Low Interest, Regulation, Lack of Consolidation/Competition: FinTechs, Banks, Non-Banks). Here, one suggests a dissimilar pressure for change that is more intense than ever before. Walter (2016, p. 29) also implies an unprecedented ever-increasing rate of environmental changes.

Technology—in accordance with Moore's Law, as introduced by Gordon Moore in 1965, includes a continuously increasing quantity of information and data, increasing storage ability, and the use of micro sensors and contemporary learning software, which consequentially lead to a continuous doubling of complexity and technological advances (Brynjolfsson and McAfee 2014, p. 53; Lanzerotti 2006). The advances of the Internet verify Moore's law. An important concept in banking today is the concept of multi-channel distribution, which can be achieved by mixing technology and service. In this context, it can be noted that the amount of branch offices has decreased since 1997, with the exception of the economic crisis of 2007: the amount of bank branches in Germany has decreased from 66,764 to

42,110 between 1997 and 2007. By 2017, there were another 10.161 branches closed, bringing the total down to 31.949 (Deutsche Bundesbank 2019, p. 104).

Customer—studies indicate that the Internet, both stationary and mobile, is playing an even more significant role for clients. The increase of the overall amount of Internet customers in Germany on the one hand and the increased mobile internet prediction on the other hand illustrates this position. In addition, the Internet use by clients is constantly growing throughout all user groups (ARD and ZDF-Onlinestudio 2016), and the general population is growing older. In 2015, Germans 14 years of age and older spent an average of 108 min per day on the Internet, whereas in 2004, they spent just 43 min per day—the behaviour of an everyday user over a decade has been more than doubled (Frees and Koch 2015, 2016).

Market—the present low interest rate period, the tough competition, and the greater regulatory effect, for example, Basel II and III for banks, are posing a growing pressure on financial intermediaries (Walter 2016, p. 31). Basel II/III cites the restricting demands and conformity to bank regulations, such as those in respect of structures of liquidity, credit, and risk (Götzl 2016, p. 5). Not only because of the laws mentioned above but also because of the absence of regulation at FinTechs, and as well of the present low-interest period, the traditional banking business strategies are under great pressure.

Competition—in strategic business management, the term 'business model' has been created since the 1990s, and it is now widely recognised (Osterwalder and Pigneur 2010). Although the issue is often addressed, science and operational practice lacks a broadly perceived and accepted definition or a general structuring approach (Burkhart et al. 2011, p. 6; Bieger and Krys 2011, p. 1; Janello 2010, p. 30). Furthermore, there are many alternatives to financial service concepts that are far out of the legal categories of banks and insurers, with space for interpreting on a literal basis, that makes it hard to define the FinTech business models universally, acceptably, and thoroughly. In fact, there is no formal registry of FinTechs, despite first conceptual efforts, which prevents the qualitative and quantitative determination of the engaged businesses. The term FinTech was established by Dorfleitner et al. (2017) in general. Likewise, Gimpel et al. (2016) recognised eight types of company structures in their empirical study of 120 company models of chiefly German FinTech start-ups in the business-to-consumer sector, namely:

1. No-money service
2. Usage-based, fee required service
3. Subscription-based, fee service
4. Bilateral, analytical service
5. Bilateral, personalised, transactional service
6. Marketplace paid by business partner
7. Personalised intermediary paid by business partner
8. Non-personalised intermediary paid by business partner

An unidentified shift in banking is quite evident, and certain banks have noticed it. They are now attempting to react to this evolving scenario by creating their own units and enterprises via accelerators and/or business incubators, so that they have a prime mover benefit from being part of the growth of innovative business models or concepts. In this context, accelerators '[...] help fledging nascent ventures. Philosophically, incubators tend to nurture nascent ventures by buffering them from the environment to give them room to grow. In contrast, whereas accelerators speed up market interactions in order to help nascent ventures adapt quickly and learn' (Cohen 2014). Practically, accelerators and incubators differ in the following key ways: duration, cohorts, business model, selection, and education, mentorship, and network development (Cohen 2014). Despite the numerous endeavours, neither a simple and prevailing business model is yet present in the banking industry, nor is there a prevailing technology or at least a policy that leads banks through current technological and strategic developments.

## 3. Theoretical Framework

### 3.1. Sustainable (Performance Indicators) in Sustainable Bank Reporting

Sustainably reporting in general is a process of information transmission of a company or organisation to the public with a specific focus on economic, environmental, and the social issues of their daily business activities. In detail, it provides the public with information about the organisation's (core) values and activities, governance approach, and exemplifies the linkage between business strategy and its engagement to global sustainability aspects. Following the Global Reporting Initiative (2019) (GRI), 'Sustainability reporting can help organisations to measure, understand and communicate their economic, environmental, social and governance performance, and then set goals, and manage change more effectively. A sustainability report is the key platform for communicating sustainability performance and impacts—whether positive or negative'.

Within the concept of sustainable reporting, performance indicators make reference to the measurements that qualitatively or quantitatively assess the sustainable performance of a company. In both inner and external control and leadership, along with outer communication, the indicators might be applied. For instance, they serve the consumers of the capital market who incorporate them into their analytical methods or use them to identify significant factors (e.g., emissions per unit of output). Though, the numerous approaches employed need to be differentiated.

### 3.2. Indicators with Regard to GRI and EFFAS

DNK reporting adopts the performance indicators chosen from the GRI and the European Federation of Financial Analysts Societies (EFFAS). They are also documented in the very same way as the DNK criteria (German Council for Sustainable Development 2012, 2019). The GRI is an ongoing global corporate reporting dialogue involving businesses and their stakeholders. The GRI creates guidelines to improve reporting quality, standardising it, and enhancing comparability. In 2016, the Sustainability Reporting Standards (SRS) were further elaborated based on the GRI G4 guiding principle. On demand for more modularity and flexibility in reporting alternatives and formats, this transformation was rendered. To complement the reporting of the DNK, a range of GRI performance indicators are used (alternatively, the 'EFFAS Key Performance Indicators'). EFFAS is also a network of European economic analysts who in 2010 published a directive for the inclusion in economic financial reporting of environmental and social elements, the 'Key Performance Indicators for Environmental Social and Governance Issues' along with the German Association for Financial Analysis and Asset Management (DVFA). Therefore, the DNK supports the 16 EFFAS indicators, in combination with the 28 performance indicators of the GRI/SRS. Performance indicators in the industry or business sector will also be taken into account. The energy consumption per ton unit, a paper consumption per worker, or the proportion of females in senior leadership are examples of performance indicators.

### 3.3. Non-Financial Performance Indicator with Regard to the German Commercial Code

The domestic requirements for the preparation of management reports are governed by § 289 of the German Commercial Code. These management reports are compelling elements of the annual report. Such an organisational report is obliged by law to be drawn up by businesses (§ 264 para. 1 German Commercial Code). A thorough assessment of the course of business and the condition of the organisation shall be provided for in the management report (§ 289 para. 1 German Commercial Code). § 289 para 3 of the German Commercial Code further asserts that major companies should also include non-financial performance indicators in the assessment related to their report in addition to financial performance reporting. More specifically, data on environmental and staff issues are included in the non-financial performance indices. The principles in this issue are laid down in § 289 para 3 of the German Commercial Code and in the group management report regulation in § 315 of the German Commercial Code, with almost identical statements developed. At the European level, in Art. 19 para. 1 of the EU Accounting Directive 2013/34/EU, regulations on non-financial

performance indices are also established with nearly the same terminology. The aim of the directive is to promote cross-border investments and enhance the union's economic statements and reports in terms of comparability and government confidence. Therefore, the aims are based on the commission's prior amendment proposition. On the other hand, the rules regulating the communication of non-financial performance indices can only be considered as a streamlined version of the provisions resulting from the reform recommended.

Essentially, it is a matter of assessing a directive that offers businesses far-reaching liberty to indicate non-financial data in the management report as far as content requirements are concerned, at both the domestic and European level. The legislature simply defines the importance (§ 289 para 3 German Commercial Code) or appropriateness (Art. 19 para. 1 EU Accounting Directive 2013/34/EU) of the data and information. However, the business must decide which details are deemed important and suitable. For instance, employee information (e.g., age structure, sick leave, and turnover) is widely recognised and should be communicated accordingly. In the context of the above-mentioned suggestion by the commission, data/information may be a guide for ecological disclosure, at least according to the dominant view in the literature (Lange 2013). Thus, although not controlled by law, there seems to be a de facto restriction.

*3.4. The Sustainability Code*

The DNK, which is stated as the transparency standard in corporate sustainability leadership, is a thorough method to sustainable reporting in Germany. A high degree of corporate transparency can only be achieved through comprehensive reporting.

The code serves as the main theoretical framework for this analysis and is designed to provide integrated financial reporting. In its practical form, it contains guidelines such as the Global Reporting Initiative or the European Federation of Societies of Financial Analysts. It is an innovation for corporate sustainability reporting both in content and in processes and brings it to a new standard level. The DNK defines specific requirements for appropriate reporting while other national and European regulations, directives, and manuals are designed to prevent a too-specific strategy.

The target groups of the sustainable reports comprise not only enterprises from all industries, but also organisations, foundations, non-governmental organisations (NGOs), trade unions, universities, science, and the media that can function in compliance with the code (Bachmann et al. 2017, p. 9). In essence, DNK implementation is voluntary and functions only as assistance for complete reporting. However, as briefly as businesses affirm the code request, it does make the transparent submission of corporate accountability mandatory. A business can use an autonomous organisation to invoke DNK with a Declaration of Conformity (Bachmann et al. 2017, p. 19). Moreover, businesses and organisations, in compliance with the 'comply or explain' principle, must state if and to what extent they satisfy the DNK requirements (Bachmann et al. 2017, p. 19). The code is considered complying with the GRI G4/SRS with its 28 outcome indices, or the EFFAS with its 16 criteria, if the report fulfils the optimal reporting norms (Bachmann et al. 2017, p. 19; Deutscher Sparkassen- und Giroverband 2017).

The DNK is organised in terms of structure by a total of 20 criteria, each of which is further clarified or described with chosen non-financial performance indices of GRI, SRS, and EFFAS, in the report domains of *Strategy*, *Process Management*, Environment, and *Society*. It is the responsibility of the respective firms to consider whether to choose GRI or EFFAS performance indicators for proficient reporting (Bachmann et al. 2017, p. 19).

Table 1 defines the 20 DNK requirements according to Bachmann et al. (2017), Bankenverband (2017), and Deutscher Sparkassen- und Giroverband (2017).

**Table 1.** Overview of DNK sustainability reporting concept.

| | | |
|---|---|---|
| **Sustainability Concept 1–10** | | |
| **Strategy 1–4** | | |
| 1. | *Strategy* | Analysis methods on opportunities and risks with regard to sustainable development. |
| 2. | *Materiality* | Sustainability strategies (e.g., strategic positioning in competition, climate-friendly/resource-saving business activity). |
| 3. | *Objectives* | Qualitative and/or quantitative and time-defined sustainability objectives, including the monitoring instruments applied to them. |
| 4. | *Depth of the Value Chain* | Information on how far down the value chain sustainability criteria are checked, and what role sustainability plays in value creation. |
| **Process Management 5–10** | | |
| 5. | *Responsibility* | Details of sustainability responsibilities in corporate governance. |
| 6. | *Rules and Processes* | Information on the rules and processes through which the sustainability strategy is implemented. |
| 7. | *Control* | Information on integrating sustainability key performance indicators into regular internal planning and control. |
| 8. | *Incentive Schemes* | Information on target agreements and compensation for executives and employees as well as information on dependencies on sustainability performance. |
| 9. | *Stakeholder Engagement* | Methods for stakeholder identification and information on regular dialogues and integration into the sustainability process. |
| 10. | *Innovation and Product Management* | Information on approaches to resource efficiency enhancement and presentation of methods for assessing and reducing the economic, environmental, and social impacts of products and services. |
| **Sustainability Aspects 11–20** | | |
| **Environmental Issues 11–13** | | |
| 11. | *Usage of Natural Resources* | Information on the extent to which natural resources are used throughout the product lifecycle. |
| 12. | *Resource Management* | Qualitative and quantitative goals in terms of increasing resource efficiency. |
| 13. | *Climate-Relevant Emissions* | Greenhouse gas emissions and targets according to the Greenhouse Gas Protocol. |
| **Society 14–20** | | |
| **Employee Matters 14–16** | | |
| 14. | *Employee Rights* | Methods of respecting workers' rights and promoting employee participation. |
| 15. | *Equal Opportunities* | Methods to promote equal opportunity, health protection, integration of migrants and people with disabilities, decent pay, family, and work arrangements as well as anti-discrimination methods of any type. |
| 16. | *Qualifications* | Methods of adapting to demographic change and measures to promote employability. |
| **Human Rights 17** | | |
| 17. | *Human Rights* | Measures to respect human rights and against forced labour, child labour, and exploitation. |
| **Social/Community 18** | | |
| 18. | *Corporate Citizenship* | Information on corporate citizenship. |
| **Compliance 19–20** | | |
| 19. | *Political Influence* | Information on participation in legislative procedures as well as information on lobby activities, payments of membership fees, payments to governments, and donations to parties/ politicians. |
| 20. | *Conduct that Complies with the Law and Policy* | Information about systems and processes to prevent unlawful behaviour (especially corruption) based on accepted standards. |

Source: Author's representation based on Bachmann et al. (2017), Bankenverband (2017), and Deutscher Sparkassen- und Giroverband (2017).

In addition, the above indicators include specific application references and terms (Bachmann et al. 2017, p. 35; Deutscher Sparkassen- und Giroverband 2017, p. 3; Bankenverband 2017). Experts originally suggested seven different variants for the implementation of the sustainability reporting requirements of a business (German Council for Sustainable Development 2012, p. 33). The alternatives for including the management report of businesses or publishing a discrete non-financial report in time or alternately creating a transparency platform on the Internet on which that report will be released are provided for legal consideration under § 289b of the German Commercial Code (Bachmann et al. 2017, p. 26).

It is noteworthy that the implementation of the DNK makes the company's sustainability reporting quite challenging. However, it is important to note that in practice, both the DNK implementation and any external assessment of sustainability reports by, for instance 'auditors/non-governmental organisations' are optional (German Council for Sustainable Development 2012). The 'comply or explain' principle (§289c para. 4 German Commercial Code) could demonstrate to be a flaw, particularly with respect to unaudited findings, since reasons for diverging from certain code requirements could not be clear. This leads to company design liberty, which reduces the depth and accuracy of reports considerably.

## 4. Methodology

This research is based on sustainable development reports that are freely accessible online through the DNK database. The database includes the selection options of the industry, business headquarters, reporting years, and the number of workers of banks, insurance companies, and financial services providers. Through this, 140 bank institutions with > 500 employees and business offices throughout Germany could be selected. A total of 135 reports for the years 2012–2017 were identified by further manual choice of banks alone; of these, n = 113 sustainability reports were finally considered for the analytical purpose, with a focus on the 2017 reporting period during which the non-financial reporting law was first applied. Before 2017, only a few reports were available, which in turn explains the rapid development of publications. The keyword analysis is methodically applied to the search for chosen terminologies within the appropriate documents in accordance with Christensen et al. (2018) and Zhao et al. (2017).

The term 'digital' is searched through the thematic focus of the work, which is coded in accordance with the 20 DNK criteria and is allocated for additional assessment in the respective reports using qualitative data analysis software MAXQDA (version 2018.02). As soon as a paragraph or sentence deals with a digital topic, the sentence is coded accordingly and assigned to the appropriate criteria. The keyword 'digital' serves as a means of identification.

On the basis of this procedure, the criteria of the DNK can be identified which are increasingly concerned with the topic of digitalization. To ensure that the analysis and codings are valid, the search is not case-sensitive and ensures that the analysis includes all references to the search term. To facilitate and clarify the (future) content analysis, when the keyword 'digital' is identified several times within a sentence, the entire sentence is coded and evaluated as one coding. Explanations for the individual variables are given in Table 1. A list of variable values and codes are given in Table 2.

Based on the individual codings, a multiple regression analysis based on the content analysis is performed, where the sum of the 'digital' codes of each sustainability report is the dependent variable and the independent variable representing the set of 'digital' codes of each DNK criterion. In this context, the high awareness of the topic is reflected in a large number of codes. Multiple regression analysis tests whether there is a relationship between several independent variables (the DNK criteria) and the dependent variable (the sum of 'digital' codes). This quantitative part is achieved via SPSS (version 25). The formula for calculating the regression function will be implemented pursuant to Shopiya (2018) and Salman and Hamid (2018).

**Table 2.** Quantitative content analysis.

| Code | Name | Files per Coding | In Percent | In Percent (Valid) | Number of Codings |
|------|------|------------------|------------|--------------------|-------------------|
| STR | 1. Strategy | 14 | 12.28 | 12.61 | 17 |
| MAR | 2. Materiality | 81 | 71.05 | 72.97 | 172 |
| OBJ | 3. Objectives | 28 | 24.56 | 25.23 | 53 |
| DVC | 4. Depth of the Value Chain | 35 | 30.70 | 31.53 | 45 |
| RO | 5. Responsibility | 7 | 6.14 | 6.31 | 9 |
| RP | 6. Rules and Processes | 12 | 10.53 | 10.81 | 16 |
| CO | 7. Control | 10 | 8.77 | 9.01 | 15 |
| IS | 8. Incentive Schemes | 0 | 0.00 | 0.00 | 0 |
| SE | 9. Stakeholder Engagement | 40 | 35.09 | 36.04 | 67 |
| IPM | 10. Innovation and Product Management | 60 | 52.63 | 54.05 | 114 |
| UNR | 11. Usage of Natural Resources | 54 | 47.37 | 48.65 | 78 |
| RM | 12. Resource Management | 40 | 35.09 | 36.04 | 63 |
| CRE | 13. Climate-Relevant Emissions | 12 | 10.53 | 10.81 | 14 |
| ER | 14. Employee Rights | 12 | 10.53 | 10.81 | 15 |
| EP | 15. Equal Opportunities | 3 | 2.63 | 2.70 | 3 |
| QU | 16. Qualifications | 15 | 13.16 | 13.51 | 22 |
| HR | 17. Human Rights | 0 | 0.00 | 0.00 | 0 |
| CC | 18. Corporate Citizenship | 7 | 6.14 | 6.31 | 7 |
| PI | 19. Political Influence | 0 | 0.00 | 0.00 | 0 |
| CCLP | 20. Conduct that Complies with the Law and Policy | 1 | 0.88 | 0.90 | 1 |
| GE | General | 7 | 6.14 | 6.31 | 10 |
| DI | Digital | 111 | 97.37 | 100.00 | 689 |
|  | Reports without Code(s) | 3 | 2.63 | - |  |
|  | Reports with Code(s) | 111 | 97.37 | 100.00 |  |
|  | Analysed Reports | 114 | 100.00 | - |  |

The standardised multiple regression function is:

$$y = \beta_1 x_1 + \beta_2 x_2 + \cdots + \beta_n x_n$$

where
| | |
|---|---|
| $y$ | is dependent variable 'digital', |
| $n$ | is number of observations, |
| $x_1, x_2, \ldots x_n$ | is independent variable, and |
| $\beta_1, \beta_2, \ldots \beta_n$ | is the regression coefficient of $x_1, x_2, \ldots x_n$. |

## 5. Results and Discussion of Findings

A total of 689 codes applicable to constant digital (DI) are found in the sustainability report evaluation for the 'digital' keyword. The reports show an imbalance in the data due to the classification of the content of the respective criteria. The Gaussian distribution for this model cannot be presumed because of the methodological approach. The respective features are shown in Table 2. The following variables for the overall model are the missing values: 8. Incentive Schemes, 17. Human Rights, and 19. Political Influence. Thus, they will be removed from the analyses.

The findings of the whole model indicate a strong coefficient of determination (r square = 0.961) and effect size (r = 0.980) in the regression analysis (Cohen 1992). In light of all legitimate criteria, the model represents a level of 96.1% of the variance of constants and only 3.9% of the deviation (see Table 3).

The application criteria that have been verified and met through the ANOVA variance analysis (see Table 4) have to be fulfilled, with a very considerable value between the respective predictors for the verification of the existing model using multiple regression analyses.

**Table 3.** Model summary.

| Model | R | R Square | Adjusted R Square | Std. Error of the Estimate |
|---|---|---|---|---|
| 1 | 0.980a | 0.961 | 0.953 | 0.853 |

**Table 4.** ANOVA.

| | Sum of Squares | df | Mean Square | F | Sig. |
|---|---|---|---|---|---|
| Regression | 1683.767 | 18 | 93.543 | 128.503 | 0.000a |
| Residual | 69.155 | 95 | 0.728 | | |
| Total | 1752.921 | 113 | | | |

The model is characterised by a small multicollinearity of the individual variables, in relation to these demands, (see Table 5—Collinearity Statistics), which lie mostly between 0.265–0.847 for Tolerance values and 1.180–3.775 for VIF values for all significant variables—with the exception of the criterion of 5. Responsibility (Tolerance 0.171/VIF 5.859), 6. Rules and Processes (Tolerance 0.316/VIF 3.167), and 7. Control (Tolerance 0.265/VIF 3.775), which varies considerably from other variables. Whereby all forms show no multicollinearity based on the following suspicion values, Tolerance < 0.1/VIF >10.

**Table 5.** Regression Summary: Coefficients and Collinearity.

| | Unstandardised Coefficients | | Standardised Coefficients | t | Sig. | Collinearity Statistics | |
|---|---|---|---|---|---|---|---|
| Code | B | Std. Error | Beta | | | Tolerance | VIF |
| (Constant) DI | 0.149 | 0.158 | | 0.947 | 0.346 | | |
| STR | 1.167 | 0.201 | 0.132 | 5.802 | 0.000 | 0.801 | 1.248 |
| MAR | 0.835 | 0.056 | 0.328 | 14.823 | 0.000 | 0.847 | 1.180 |
| OBJ | 0.950 | 0.091 | 0.232 | 10.465 | 0.000 | 0.847 | 1.181 |
| DVC | 1.082 | 0.130 | 0.188 | 8.326 | 0.000 | 0.810 | 1.234 |
| RO | −0.348 | 0.589 | −0.029 | −0.590 | 0.556 | 0.171 | 5.859 |
| RP | 0.977 | 0.326 | 0.109 | 3.001 | 0.003 | 0.316 | 3.167 |
| CO | 0.842 | 0.345 | 0.097 | 2.438 | 0.017 | 0.265 | 3.775 |
| IS [1] | | | | | | | |
| SE | 1.070 | 0.090 | 0.287 | 11.880 | 0.000 | 0.714 | 1.401 |
| IPM | 1.043 | 0.064 | 0.379 | 16.201 | 0.000 | 0.758 | 1.320 |
| UNR | 0.834 | 0.094 | 0.202 | 8.906 | 0.000 | 0.809 | 1.237 |
| RM | 0.865 | 0.094 | 0.211 | 9.167 | 0.000 | 0.784 | 1.275 |
| CRE | 0.693 | 0.236 | 0.067 | 2.934 | 0.004 | 0.802 | 1.248 |
| ER | 1.169 | 0.215 | 0.128 | 5.440 | 0.000 | 0.750 | 1.334 |
| EP | 1.481 | 0.567 | 0.060 | 2.610 | 0.011 | 0.774 | 1.292 |
| QU | 0.900 | 0.166 | 0.129 | 5.418 | 0.000 | 0.738 | 1.356 |
| HR [1] | | | | | | | |
| CC | 0.899 | 0.362 | 0.055 | 2.483 | 0.015 | 0.845 | 1.184 |
| PI [1] | | | | | | | |
| CCLP | 0.831 | 0.933 | 0.020 | 0.891 | 0.375 | 0.844 | 1.184 |
| GE | 0.756 | 0.193 | 0.083 | 3.914 | 0.000 | 0.924 | 1.083 |

[1] Cannot be computed because at least one of the variables is constant.

With regard to the previously mentioned research questions one and two:

RQ1: The level of understanding of the respective criteria, regarding the notion of digital sustainability, shows that banking mainly reports on the criteria of 2. Materiality (172 codes), 3. Objectives (53 codes), 4. Depth of the Value Chain (45 codes), 9. Stakeholder Engagement (67 Codes), 10. Innovation and Product Management (114 codes), and 11. Usage of Natural Resources (78 codes),

as well as 12. Resource Management (63 codes)—see Table 2. It can be noted that the issue of digitalization is neglected and receives little or no attention compared to all previous unnamed criteria.

RQ2: Table 6 shows the Correlation of Pearson (PCC) on the 'digital' constant. All predictor values indicate between very high significance levels of 0.01 and 0.05. Only criteria 1. Strategy, 15. Equal Opportunities, 18. Corporate Citizenship, and General do not confirm to this level. There are systematically positive linear relationships, but there are no strong uphill (positive) linear relationships (>0.7). The strongest PCCs (0.531 and 0.574) are measurable for the DNK Criteria of 9. Stakeholder Engagement and 10. Innovation and Product Management. These are followed by criteria 11. Usage of Natural Resources (0.441), 12. Resource Management (0.402), and 2. Materiality (0.373). While the initial keyword analysis reveals that criterion 2. Materiality is highly dominant, the findings of the regression analysis confirms that impact (PCC 0.373/stand. $\beta$ 0.328). However, the previous dominant position of 10. Innovation and Product Management, is reconfirmed with a PCC of 0.574 and a $\beta$ of 0.379 and leads to the conclusion that 2. Materiality has slightly less influence on 'digital'. Both are followed by criteria 9. Stakeholder Engagement ($\beta$ 0.287), 3. Objectives ($\beta$ 0.232), and 11. Usage of Natural Resources ($\beta$ 0.202).

**Table 6.** Regression summary—correlations.

| | Pearson Correlation | Sig. (1-Tailed) |
|---|---|---|
| Code | Digital | |
| (Constant) DI | 1000 | |
| STR | 0.133 | 0.079 |
| MAR | 0.373 ** | 0.000 |
| OBJ | 0.345 ** | 0.000 |
| DVC | 0.213 * | 0.011 |
| RO | 0.199 * | 0.017 |
| RP | 0.222 ** | 0.009 |
| CO | 0.350 ** | 0.000 |
| IS [c] | | |
| SE | 0.513 ** | 0.000 |
| IPM | 0.574 ** | 0.000 |
| UNR | 0.441 ** | 0.000 |
| RM | 0.402 ** | 0.000 |
| CRE | 0.168 * | 0.037 |
| ER | 0.320 ** | 0.000 |
| EP | 0.071 | 0.226 |
| QU | 0.338 ** | 0.000 |
| HR [c] | | |
| CC | 0.058 | 0.271 |
| PI [c] | | |
| CCLP | 0.169 * | 0.037 |
| GE | 0.043 | 0.325 |

[c] Cannot be computed because at least one of the variables is constant. ** Correlation is significant at the 0.01 level (1-tailed). * Correlation is significant at the 0.05 level (1-tailed).

## 6. Conclusions and Further Research

This study aggregated report evidence on the digitalization relationship in sustainable bank reports. It was also directed to uncover whether all DNK standards are adequately taken into account within each report in order to illustrate the extent to which banks are currently addressing the issue of digitalization. As the banking market is changing and new business models and technologies are transforming the current market and disrupting existing business models, addressing the issue of digitalization is essential for stakeholders to keep them fully informed about the evolution and current situation. Holistic reporting is essential in this context. This paper highlights a conceptual gap and imbalance in an existing reporting approach based on the results of the study. In this context, the model presented in this paper expresses the statistical dependence between digitalization and a variety of input

variables, namely the DNK criteria. Given the importance of the issue, it should be given equal priority in all reporting approaches. However, the results show that the issue of digitalization is only partially addressed in banks' reporting and is not fully and equally reflected, which in turn leads to an incomplete and insufficient transfer of information to the market and stakeholders. This recognition of the thematic imbalance is important for practice and science, as managers are sensitised to shortcomings in the area of transparent reporting and thus can adapt their actions accordingly. Furthermore, science has the task of reacting to these circumstances and can thus further develop existing reporting approaches.

The results of the analysis are particularly revealing as they uncover the different criteria included in the digital reporting obligations, which have never been analysed before. This work suggests that the banking sector is particularly focussed on strategic areas in terms of materiality and objectives as well as process management levels in terms of stakeholder engagement, innovation and product management, and environmental issues. Our findings indicate that social issues are poorly or not taken into account at all in the DNK framework and in the context of digitalization; therefore, an important social as well as corresponding strategic thought is not considered. The findings can be explained by the literature, suggesting that there is an obvious need for banks, companies, and managers to adapt effectively to the rapidly changing markets in terms of technology, customers, and the competition, but that information about this process of change becomes less important.

Furthermore, the investigation leads to the conclusion that today's reporting is not fully suitable for information purposes. Using the example of reporting on the subject of digitalization, it can be deduced that topics are not given equal consideration and banks are left with too much leeway and freedom within the formulation of these reports. This, in turn, could lead to the neglect of important issues, which could entail the withholding of information. The further development of these standards is unavoidable; financial institutions should report on the same basis in order to ensure a holistic and transparent flow of information.

A few potential limitations of this study have to be addressed. This meta-analysis attempted to avoid a term bias by a non-sensitive analysis, providing that all references to the term 'digital' are included. Yet, access to such database work is limited. The keyword analysis used to search the chosen terminologies within the appropriate reports should probably contain more words or phrases to ensure the effectiveness of the content analysis. Another limitation relates to the sample of the studies included. We analysed sustainable reports coming from 140 banks, insurance companies, and financial services providers with >500 employees and business offices throughout Germany. A more homogenous future approach could also include small or medium firms and, thus, a substantial variance in firms' sizes can be observed. In order to identify the consequence of resource scarcity more clearly, the sampled study could have been broadened to small firm studies.

Our study provides some directions for future research. An important imbalance was identified in the context of the 'digital topic' and what precipitates it. Hence, more research is needed to consider and analyse new approaches that aim to further develop existing reporting standards that make the comprehensive reporting of important subjects mandatory and applicable. Moreover, while we presented various explanations as if banks do make sustainable reports, future studies should examine particularly why certain criteria are reported or not and look at the managerial factors justifying this. An improved understanding and interpretation of the criteria and the use of more precise specifications would enable the stakeholders to reach a greater degree of significant and equivalent reporting and restrict content-related individuality. Furthermore, we know little about the company structures and geographical delineations of business models. Future analyses should also be carried out at a deeper level of content, so that the keyword analysis is not only limited to the framework, but also includes interpretations of the content of the respective reporting.

Overall, this study identified a number of important contextual criteria that banks focus most on in their sustainability reports. In so doing, we hope to foster a more contextual understanding of the field of the sustainability reporting in banking with regard to digitalization. The main indicators for a variety of criteria have been identified, but we do not want to claim that the identified criteria

are the only ones used for analysis. Further research may focus on uncovering other standards and illustrating specific reports on sustainability in the banking sector to support banks and managers in achieving full and balanced banking reporting.

**Author Contributions:** Conceptualization, F.D.; Investigation, F.D.; Methodology, F.D.; Project administration, M.Š.; Validation, F.D.; Writing—original draft, F.D.; Writing—review & editing, F.D. All authors have read and agreed to the published version of the manuscript.

**Funding:** This research received no external funding.

**Conflicts of Interest:** The authors declare no conflict of interest.

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
