# Peer review of "The Role of ‘Digitalization’ in German Sustainability Bank Reporting"

_ijfs, doi:10.3390/ijfs8010016_

Round 1
Reviewer 1 Report
The paper contains new and significant information by incorporating the importance of sustainable reporting and the lack of the megatrend digitalization with a keen eye on the banking industry.
The relevant literature in the field of sustainability and digitalization gives a complete understanding of the actual relevance. As well as the development, the status quo and the challenges in a disrupting century between the old economy driven by the quickly evolving and dynamic business models in the new economy.
The paper´s argument is built on an appropriate base of theory and its influence on the society regarding the systematic role of the banking industry in the economy and the sustainable change. The Article address the impact in the financial industry and especially the core business of the traditional banking industry.
The Article adresess the impact in the financial industry and more generally of the core business from the traditional banking which will be influenced by the evolution of digitalization and sustainability in the 21th Century. The Author deals with a sophisticated and clear argumentation structure and thus enables the reader to reveal the link between the digitalization and sustainability in the banking sector.
The theoretical approach, scientific methodology and results in the paper are well applied.
The paper bridges the gap between the theoretical discussions of the influences by digitalization with regard to the sustainable development and the important application from the perspective of the banking industry itself. From the perspective of the different stakeholders it is really important to measure the risk by incorporating the non-financial facts and figures in the sustainability reporting.
I wish to read some more about the implications of the results to the society and the impact of the stakeholders mentioned in the paper. I recommend that the author should discuss the importance of sustainability reporting in greater detail at the very end of the paper. This would once again underline the holistic character of the topic for the reader.
Overall the results presented clearly and the analysis of the two research questions is appropriately. The Author express its research questions and following analysis clearly in the paper and fulfilled the expected knowledge and technical language of the journals readership. The summary addresses the research questions and clearified the questions adequately.
It is adequate to justify publications.
Author Response
Dear reviewer,
we thank you very much for your efforts and the review of our article.
With regard to your comments, please allow us to explain as follows:
We have taken your comments into account and expanded both the introduction and the conclusion of the article so that a more detailed description of the topic is provided for the reader. Furthermore, we have once again concretized the methodology in order to achieve a higher level of understanding. We had the article proofread by a native English speaker.For the sake of clarity, we have marked the changes made in yellow.
With thank you again for your effort.

Reviewer 2 Report
The paper focuses on the German banking sector and is trying to analyse the connections between digitalisation transition process and banks sustainable reporting but there are parts needing clarifications and scientific arguments to sustain the followings:
a) the relation between sustainability reporting and banks digitalisation.
b) the relation between the German sustainability code and banks digitisation.
c) explain the meaning of digital sustainability,
d) the methodology is not clearly explained, concerning the variables used.
( provide a list with variable values, and an example of codification)
e) the relevance of this study for different categories of readers.
f) the paper contributes to the literature
Author Response
Dear reviewer,
we thank you very much for your efforts and the review of our article.
With regard to your comments, please allow us to explain as follows:
a) We have taken up this approach and extended 1. introduction by some passages. We have considered the importance of holistic reporting once again and briefly discussed the connection between technological development and the changing financial market.
b) In the introduction, we discussed the consequences of not adapting to technological developments and addressed the danger of disrupting traditional business models in the banking sector once again. In this context we point out the importance of comprehensive reporting. This is to underline the importance of the topic once again.
c) In the penultimate section of the introduction we took up your idea of sustainable digitalisation and addressed the approaches of Stuermer et al. (2017) We have transferred their findings to the context and proceed with the RQ.
d) We clarified the methodology again and completed the coding process. We also provided cross-references to tables 1 and 2. These explain the Independent Variables and their abbreviations and values, which is also the reason for not presenting the variables in more detail in the Methodology section.
e) We have considered your comments and added the points you raised to the last part of the article.
f) Should this also be supplemented? Based on the wording, we assume that no further changes are necessary.
Regarding the English language and style:
We had the article proofread by a native English speaker.
For the sake of clarity, we have marked the changes made in yellow.
With thank you again for your effort.
Round 2
Reviewer 1 Report
Independent Review (v2)
Journal: International Journal of Financial Studies
Manuscript ID: ijfs-671290
Type of manuscript: Article
Authors: Florian Diener, Miroslav Spacek*
Title: Sustainability Reporting in Banking: The Awareness of Digitalisation in Bank Reporting with regard to the German Sustainability Code
Comments by the reviewer
The paper contains new and significant information by incorporating the importance of sustainable reporting and the lack of the megatrend digitalization with a keen eye on the banking industry.
The relevant literature in the field of sustainability and digitalization gives a complete understanding of the actual relevance. As well as the development, the status quo and the challenges in a disrupting century between the old economy driven by the quickly evolving and dynamic business models in the new economy.
The paper´s argument is built on an appropriate base of theory and its influence on the society regarding the systematic role of the banking industry in the economy and the sustainable change. The Article address the impact in the financial industry and especially the core business of the traditional banking industry.
The Article adresess the impact in the financial industry and more generally of the core business from the traditional banking which will be influenced by the evolution of digitalization and sustainability in the 21th Century. The Author deals with a sophisticated and clear argumentation structure and thus enables the reader to reveal the link between the digitalization and sustainability in the banking sector.
The theoretical approach, scientific methodology and results in the paper are well applied.
The paper bridges the gap between the theoretical discussions of the influences by digitalization with regard to the sustainable development and the important application from the perspective of the banking industry itself. From the perspective of the different stakeholders it is really important to measure the risk by incorporating the non-financial facts and figures in the sustainability reporting.
Overall the results presented clearly and the analysis of the two research questions is appropriately. The Author express its research questions and following analysis clearly in the paper and fulfilled the expected knowledge and technical language of the journals readership. The summary addresses the research questions and clearified the questions adequately.
The second version is adequate to justify publications.
Author Response
We thank the reviewer for his time and for his evaluation.
Reviewer 2 Report
There are improvements but still, many issues are not at all clear. It is a mixture of two concepts digitalisation and sustainability reporting, using the financial sector case, but the link between the elements are missing, and the paper is hard to be understood. Here below are some of my concerns.
The title is too long, and this creates ambiguity.2. In the German Sustainability Code, there are requirements related to "digital" issues? If yes explain it, if no for what reason to be analysed?
3. I suppose you analyse the content, the requirements not the structure of GSC. For what reason might be relevant to the structure of GSC?
4 In Introduction insert resources cited to support your phrases like for ex: With regard to the banking crisis in 2007, there was a lack of sufficient and holistic reporting and regulatory standards in the context of the banking system .....
5. you mention that digital sustainability "refers exclusively to intangible goods" so why you search after the word "digital"?
6 Explain exactly what you are mean by digital sustainability in the financial sector.
7. Which were your expectations to find in the banks' sustainability reporting, in relation to the germen context?
8 Why german context is relevant?
Author Response
We thank the reviewer. In the following you will find our statements:
1. The title is too long, and this creates ambiguity.
We have shortened the title and are focusing on the analysis approach (the DNK) and our thematic main focus (Awareness of Digitisation).
Title suggestions:
The Role of 'Digitalisation' in German Sustainability Bank Reporting
2. In the German Sustainability Code, there are requirements related to "digital" issues? If yes explain it, if no for what reason to be analysed?
In the DNK we cannot find any obligations for banks or companies to deal with specific issues. The scope and detailed treatment of the topics is at their own discretion. It is merely a framework which the banks can follow and which can serve as a guide. Only “individual services or non-financial aspects, including environmental, employment and social issues, staff, respect for human rights, battle against corruption and diversity, must be reported to the governing agencies by the businesses involved, however, the main areas of focus are not defined more precisely.“
Digitalisation is a key issue that should be mentioned in every report and respective categories, especially as it is a fundamental topic for the financial sector and its customers. “On the basis of 26 interviews we conducted with German bank managers and specialists, it became evident that digitalisation is a key issue for banks, which should be approached at different levels and from different perspectives. The results of the interviews identified it as an elementary component of current and future banking business.” In the introduction, we first outlined the context and importance of digitalisation and digital sustainability as well as the stakeholders' interest in information and then derived the two research questions.
3. I suppose you analyse the content, the requirements not the structure of GSC. For what reason might be relevant to the structure of GSC?
As digital content (keyword analysis) is at the heart of our paper, we have explained why and for whom DNK is important and what it focuses on: “The aim of DNK is to ensure that medium-sized banks can apply this standard as an easy-to-use instrument, as banks will be obliged to report on non-financial elements of their actions due to the large number of employees. […] Since digitalisation is increasingly influencing the entire economic and market environment, the financial market is also affected by this change and must address and position itself in the context of its importance and duty to inform its customers and the market. […] the DNK provides the basis for a holistic view of current developments in the financial market.”
4 In Introduction insert resources cited to support your phrases like for ex: With regard to the banking crisis in 2007, there was a lack of sufficient and holistic reporting and regulatory standards in the context of the banking system …
We have quoted a highly respected source from the European Accounting Review (Barth, M. E., & Landsman, W. R. (2010). How did Financial Reporting Contribute to the Financial Crisis? European Accounting Review, 19(3), 399–423. doi:10.1080/09638180.2010.498619), which refers to the lack of transparency and the impact of financial reporting in times of crisis in 2007 and also describes the thematic context.
5. you mention that digital sustainability "refers exclusively to intangible goods" so why you search after the word "digital"?
Digitalisation is the further development on a technological level. Digitalisation is not tangible – It is a process that is based on knowledge, which, in turn, leads to changes on an operational level and should be addressed for the reasons of market and technological development. Even if knowledge is not tangible, it plays an essential role.
6 Explain exactly what you are mean by digital sustainability in the financial sector.
In lines 108-134 we discuss the topic of digital sustainability and the three main criteria of digital sustainability in more detail and place them in the context of financial services.
7. Which were your expectations to find in the banks' sustainability reporting, in relation to the germen context?
As digitalisation is increasingly having an impact on several levels within a bank and past and future developments have a significant impact on the financial market, it can be expected that banks address important issues and report on them in detail. “In order to satisfy the information needs of stakeholders, banks are now addressing the various problems of a sustainable economy in a variety of ways. Social issues, market developments, as well as technological topics like digitalisation, which are of great importance for banks today, will be discussed.” In particular due to the lack of transparency in reporting in recent years and todays constant digital market development, banks receive official support in terms of the DNK framework. We are concerned whether the banks are addressing this important issue and expected that they have not sufficiently addressed the issue of digitalisation. This assumption was corroborated by our results. We also assumed that the banks did not report adequately and in an appropriate format, which was also confirmed. This leads to research questions 1 and 2.
8 Why german context is relevant?
“As a result of the forthcoming Brexit, the German banking system will become increasingly important as, in addition to the French financial sector, it is an important pillar for the European system (Howarth and Quaglia 2018). The high German standards can be seen as an example of high efficiency and reliability and thus as an outstanding example for a banking system that is still in development. Furthermore, this system finances the strongest economy in the EU and is considered as one of the strongest in the world.” “Even though the DNK is primarily a reporting approach for the German market, the insights already gained can serve as a guideline and can be transferred to respective markets in other countries and their legislation, so that cross-national comparability can be achieved based on a holistic approach.” “Due to the accessibility, the available number of reports, and the broad acceptance of the DNK, this paper focuses in particular on the application of the framework in the German banking sector.”

Round 3
Reviewer 2 Report
Thanks for the response but the paper is still confused.
I agree with the idea of digitalisation and digital sustainability but, if in the DNK are no obligations to deal with specific issues related to digitalisation, your presumptions and expectations are wrong, especially in the cultural context of Germany, based on rules, regulations.
In the abstract, you mentioned "It concerns the assessment of mandatory sustainable reporting .........and stricter legal requirements for stakeholder
data responsibility.
So is more related to voluntary reporting of issues related to "digital".
Also if we discuss about your search, this was limited to word "digital" but there are many other words that might be used to express the idea.
So why to search for something about which there are no references in the DNK, when from the beginning the response is clearly negative, confirmed also by your results.
"The paper shows a structural inequality within sustainable bank reporting with regard to digitalisation. It also shows that issues are not adequately addressed and covered in legal reporting standards and that the provision of
information to stakeholders on specific issues is largely undefined'
Coming back to the previous Q3. I suppose you analyse the content, the requirements not the structure of GSC. For what reason might be relevant to the structure of GSC? I reformulate the question as follows; you analysed the requirements' content or the structure of GSC?
Author Response
We thank the reviewer for his effort. In the following we will go into the respective comments in detail.
I agree with the idea of digitalisation and digital sustainability but, if in the DNK are no obligations to deal with specific issues related to digitalisation, your presumptions and expectations are wrong, especially in the cultural context of Germany, based on rules, regulations.
The DNK is intended to address topics that are important for the transparency of information. This includes, based on historical and current developments, the topic of digitalisation. Our findings confirm that the topic is addressed within reporting, but it also shows an imbalance and insufficient focus. ‘On the basis of 26 interviews we conducted with German bank managers and specialists, it became evident that digitalisation is a key issue for banks, which should be approached at different levels and from different perspectives. Even if reporting on digitalisation in sustainability reports is voluntary, the interviews indicated that the topic was important for banks and should be addressed holistically. So the results of the interviews identified it as an elementary component of current and future banking business.’
In the abstract, you mentioned "It concerns the assessment of mandatory sustainable reporting .........and stricter legal requirements for stakeholder data responsibility. So it is more related to voluntary reporting of issues related to "digital".
As mentioned in the introduction, sustainability reporting is required from companies with more than 500 employees. Reporting in the sense of a sustainability report is mandatory by law, but the content approaches are not. However, they should represent a holistic view of the company and current (market) challenges.
It is a legal regulation respectively obligation which is valid on a European level [(Directives 2014/95/EU 2014) - see also sections 289b et seq. German Commercial Code]. Since the DNK is merely a framework for exactly this kind of reporting, the work aims to analyse the content of an important topic for the banking world and its stakeholders: i.e. digitalisation.
Also if we discuss about your search, this was limited to word "digital" but there are many other words that might be used to express the idea.
We completely agree with you. The subject of digitalisation is a very broad field and must be considered from different perspectives. In the penultimate section of the last chapter, we address this "problem" again and point out further necessary investigations in the context. 'Digital' is the truncation of the word 'digitalisation/digitalization' and supports the analysis of word bases with identical content. It also helps to identify the topic in its entirety; however, a further content analysis (further research) is of course important in this context.
So why to search for something about which there are no references in the DNK, when from the beginning the response is clearly negative, confirmed also by your results.
Even if there are no substantive rules for each DNK criterion rather recommendations, it does not mean that the issue of digitalisation should/can be viewed negatively from the outset. Based on interviews, bank managers confirmed high importance of the digital topic nowadays. We found digitalisation in almost every bank reporting; this underlines the importance once again and shows that banks are already addressing the issue. Furthermore, the results do not show that the topic of digitalisation is not addressed at all – please see 5. Results and Discussion of Findings.
The framework does not indicate a clear negative assessment of a particular topic. As important topics are to be addressed in this form of reporting, attention should also be paid to the topic of digitalisation, as it changes the banking world in a disruptive way. This is precisely our research approach.
"The paper shows a structural inequality within sustainable bank reporting with regard to digitalisation. It also shows that issues are not adequately addressed and covered in legal reporting standards and that the provision of information to stakeholders on specific issues is largely undefined'
Coming back to the previous Q3. I suppose you analyse the content, the requirements not the structure of GSC. For what reason might be relevant to the structure of GSC? I reformulate the question as follows; you analysed the requirements' content or the structure of GSC?
With the keyword analysis carried out, we focus on the presence of the topic of digitalisation. Preferably, a content-related component is considered, but the structural component was addressed in lesser extent. The focus is on the content-related aspects of digitalisation and on the focus of a specific topic within the DNK criteria.